

# Deciphering the microRNA transcriptome of skeletal muscle during porcine development

Miaomiao Mai[1,*], Long Jin[1,*], Shilin Tian[2], Rui Liu[1], Wenyao Huang[1], Qianzi Tang[1], Jideng Ma[1], An'an Jiang[1], Xun Wang[1], Yaodong Hu[1], Dawei Wang[2], Zhi Jiang[2], Mingzhou Li[1], Chaowei Zhou[3] and Xuewei Li[1]

[1] College of Animal Science and Technology, Sichuan Agricultural University, Ya'an, People's Republic of China
[2] Novogene Bioinformatics Institute, Beijing, People's Republic of China
[3] Department of Aquaculture, Southwest University at Rongchang, Chongqing, People's Republic of China
[*] These authors contributed equally to this work.

## ABSTRACT

MicroRNAs (miRNAs) play critical roles in many important biological processes, such as growth and development in mammals. Various studies of porcine muscle development have mainly focused on identifying miRNAs that are important for fetal and adult muscle development; however, little is known about the role of miRNAs in middle-aged muscle development. Here, we present a comprehensive investigation of miRNA transcriptomes across five porcine muscle development stages, including one prenatal and four postnatal stages. We identified 404 known porcine miRNAs, 118 novel miRNAs, and 101 miRNAs that are conserved in other mammals. A set of universally abundant miRNAs was found across the distinct muscle development stages. This set of miRNAs may play important housekeeping roles that are involved in myogenesis. A short time-series expression miner analysis indicated significant variations in miRNA expression across distinct muscle development stages. We also found enhanced differentiation- and morphogenesis-related miRNA levels in the embryonic stage; conversely, apoptosis-related miRNA levels increased relatively later in muscle development. These results provide integral insight into miRNA function throughout pig muscle development stages. Our findings will promote further development of the pig as a model organism for human age-related muscle disease research.

## INTRODUCTION

Skeletal muscle makes up approximately 40% of mammalian body mass (*Güller & Russell, 2010*), and maintaining skeletal muscle function is a prerequisite for maintaining normal body function. Therefore, understanding how skeletal muscle grows is important for understanding the growth of the body as a whole. Numerous conditions, including neuro-muscular disorders, sedentary lifestyles, chronic disease, and aging, have been shown to be associated with loss of skeletal muscle mass and function (*Di Giovanni et al., 2004*; *Dogra et al., 2007*; *Jagoe & Goldberg, 2001*; *Lynch, 2001*). The growth of skeletal muscle mass, like

Corresponding author
Chaowei Zhou, zcwlzq666@163.com
Xuewei Li, xuewei.li@sicau.edu.cn

the mass of any other tissue, depends on protein turnover (synthesis, degradation, and repair capacities) and cell turnover (differentiation and proliferation) (*Sartorelli & Fulco, 2004*). Cellular turnover plays a major role in embryonic muscle development. Moreover, postnatal muscle development is mainly associated with accumulation of myonuclei (satellite cell proliferation) and muscle-specific proteins (*Rehfeldt et al., 2000*). Conversely, the decrease in muscle mass, strength, and rate of contraction is closely related to aging, and these age-related changes have been observed in many species, including humans (*Evans & Lexell, 1995*), rats (*Daw, Starnes & White, 1988*; *Hooper, 1981*), and mice (*Brooks & Faulkner, 1988*). Notably, it is well documented that the remarkable structural and functional changes in skeletal muscle that occur during aging, including reduction in muscle mass and increased apoptosis (*Dirks & Leeuwenburgh, 2002*; *Janssen et al., 2000*; *Jemal et al., 2008*; *Nair, 2005*; *Navarro, Lopez-Cepero & Sánchez, 2001*; *Pollack et al., 2002*), always start at the mid-point of the lifespan (*Nair, 2005*).

MicroRNAs (miRNAs) are endogenous small non-coding RNAs that modulate gene expression at a post-transcriptional level by binding to the $3'$ untranslated region ($3'$-UTR) of target mRNAs (*Nelson et al., 2003*). To date, numerous miRNAs have been shown to be associated with skeletal muscle aging and act as regulators of cellular senescence at the tissue or organism level. For example, *Hamrick et al., (2010)* reported that miR-7, miR-468, miR-542, and miR-698 levels in mouse muscle tissue substantially increased with age, whereas miR-124a, miR-181a, miR-221, miR-382, miR-434, and miR-455 levels substantially decreased with age. Similar findings have also been found in humans, in which 18 miRNAs were differentially expressed between younger and older men (*Drummond et al., 2011*). However, these findings are not sufficient to comprehensively understand the relationship between miRNAs and muscle development, and further research is required.

The pig (*Sus scrofa*) harbors tremendous biomedical importance as a model organism, because of its closer phylogenic relationship and more similar physiology to humans than mice or other mammals (*Lunney, 2007*). Although studies have identified changes in miRNA expression during muscle development in pigs (*Chen et al., 2012*; *Hou et al., 2012*; *Huang et al., 2008*; *Li et al., 2012*; *Xie et al., 2011*; *Zhou et al., 2013*), only a small subset of ages was analyzed. Moreover, these studies mainly focused on understanding miRNAs involved in fetal and adult muscle development. Until now, little was known about the role of miRNAs in middle-aged muscle development.

Here, to identify miRNAs and determine their role in skeletal muscle development, we evaluated miRNA transcriptome profiles at five stages of muscle development, including one prenatal stage (90-d post-gestation fetuses, E90) and four postnatal stages (0 d, 30 d, 180 d, and 7 y after birth), which represented prenatal, natal, weaning, young adult, and middle-aged stages, respectively (*Bhathena, Berlin & Johnson, 1996*). Our study will contribute to understanding the underlying mechanisms of muscle development and muscular atrophy.

## METHODS

### Animal ethics statement

All animals used in this study were farmed according to the Regulations for the Adminis-tration of Affairs Concerning Experimental Animals (Ministry of Science and Technology, China, revised in June 2004) and approved by the Institutional Animal Care and Use Committee in the College of Animal Science and Technology, Sichuan Agricultural University, Sichuan, China under permit No. DKY-B20121406.

### Preparation of experimental animals and tissues

Fifteen female Jinhua pigs, which were grouped into five stages (three replicates for each stage): prenatal embryonic day 90 (E90), postnatal 0 d (natal), 30 d (weaning), 180 d (young adult), and 7 y (middle-aged), were humanely sacrificed. The *longissimus dorsi* muscle tissues from the same anatomical location were collected from pigs representing each of the five stages. Subsequently, the samples were immediately frozen in liquid nitrogen and stored at −80 °C until RNA extraction. Total RNA was isolated from each sample using Trizol (Takara) according to the manufacturer's protocol. The concentra-tion and purity of RNA samples were determined by measuring the A260/A280 ratio using a NanoDrop ND1000 spectrophotometer (Thermo Scientific, Hudson, NH, USA).

### Myofiber histology

After sacrifice, all muscle tissues were fixed in 10% neutral buffered formalin solution, embedded in paraffin using a TP1020 semi-enclosed tissue processor (Leica, Wetzlar, Germany), sliced at a thickness of 6 μm using RM2135 rotary microtome (Leica, Wetzlar, Germany), and stained with hematoxylin and eosin (H&E). The mean diameter of muscle cells was calculated as the geometric average of the maximum and minimum diameter, and 100 cells were measured for each sample in randomly selected fields using a TE2000 fluorescence microscope (Nikon, Tokyo, Japan) and Image Pro-Plus 7.0 software.

### Small RNA library construction and high-throughput sequencing

We pooled the total RNA for replicates in the E90, 0-d, and 30-d stages. For 180-d and 7-y stages, total RNA of each replicate was individually used for library construction. Therefore, a total of nine libraries were constructed and sequenced using single-end sequencing in 36 nt reads by an Illumina Genome Analyzer II. The bioinformatics pipeline for miRNA discovery and profiling was carried out as previously described, with some improvements (*Gambardella et al., 2010*). All reads were counted and the identical reads were combined into a single kind. After eliminating adaptor sequences, low-quality tags, sequences smaller than 16 bp, and reads with no insertion, all of the clean tags were annotated and classified by comparison with the non-coding RNAs (rRNA, tRNA, scRNA, snRNA, and snoRNA) in the GenBank (http://www.ncbi.nlm.nih.gov/) and Rfam 9.1 (http://rfam.sanger.ac.uk/) databases. Known miRNAs were identified by comparing our clean tags to mature miRNAs in miRBase 19.0. The remaining non-annotated sRNA sequences were aligned against the porcine genome, and genomic sequences containing the sRNA with 80–100 nt of flanking sequences were used to predict

hairpin structures using Mfold (http://mfold.rna.albany.edu). Only the sequences that could be folded into typical hairpin structures and were located in intergenic regions or introns were considered to be miRNA precursor loci of potential novel miRNAs in the porcine genome.

## miRNA differential expression and clustering analyses

IDEG6 (*Romualdi et al., 2003*) was employed to detect DE miRNAs between two libraries. A unique miRNA was considered DE when $P < 0.001$ was yielded by three statistical tests (Greller and Tobin, R of Stekel and Falciani, and general chi-squared tests) with Bonferroni correction. Hierarchical clustering analysis (HCL) of DE miRNAs was performed with MultiExperiment Viewer (MeV) (*Howe et al., 2010*).

## Prediction and functional annotation of miRNA target genes

The potential targets of a certain miRNA were predicted by PicTar (*Krek et al., 2005*) and TargetScan Human 6.2 (*Lewis, Burge & Bartel, 2005*), and the pairwise overlaps of the results from both programs composed the final predicted targets. The predictions were based on human mRNA–miRNA interactions, because of the absence of porcine miRNAs in the current version of abovementioned algorithm. The enriched GO biological process (GO-BP) and Kyoto Encyclopedia of Genes and Genomes (KEGG) pathway terms of predicted target genes were determined using the DAVID bioinformatics resource (*Huang, Sherman & Lempicki, 2008*).

## STEM analysis

STEM analysis was used to visualize expression patterns for DE miRNAs. Each miRNA was assigned to the model profile that its time series most closely matched based on the correlation coefficient. The number of miRNAs assigned to each model profile was then computed. The number of miRNAs expected to be assigned to a profile was estimated by randomly permuting the original time point values, renormalizing the miRNA expression values, assigning miRNAs to their most closely matching model profiles, and repeating this process for a large number of permutations. The average number of miRNAs assigned to a model profile over all permutations was used as the estimate of the expected number of miRNAs assigned to the profile. The statistical significance of the number of miRNAs assigned to each profile compared with the expected number was also computed (*Ernst & Bar-Joseph, 2006*).

## qRT-PCR validation

The expression changes of nine selected miRNAs were validated using an EvaGreen-based High-Specificity miRNA qRT-PCR Detection Kit (Stratagene, La Jolla, CA, USA) on the CFX96$^{TM}$ Real-Time PCR Detection System (Bio-Rad, Hercules, CA, USA). The q-PCR validations were carried out on three biological replicates. Information on the primer pairs used is available in Table S1. Three endogenous control genes (U6 snRNA, 18S rRNA, and 5S rRNA) were used in this assay. The $\Delta\Delta Ct$ method was used to determine the expression level differences between surveyed samples.

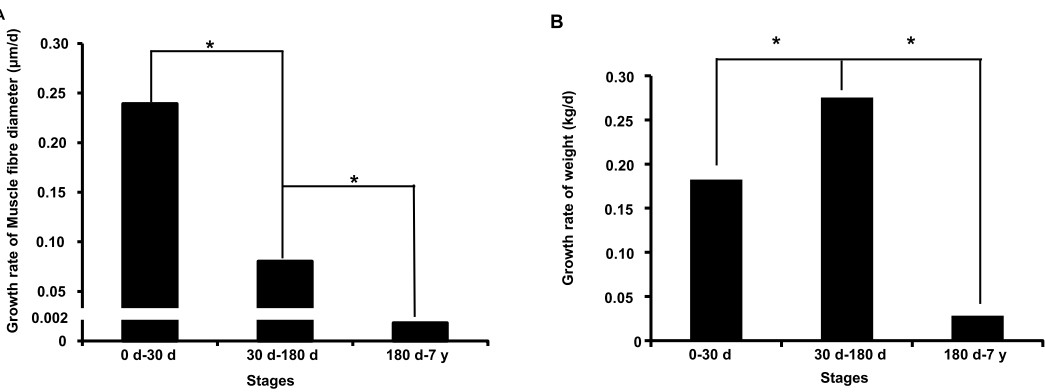

**Figure 1 Growth rate of muscle fiber diameter (A) and weight (B) during the postnatal muscle development stage.** Three replicates were used when performed the statistical analysis between different development stages. "*" indicates significant difference (Student's *t*-test, $P < 0.05$).

## RESULTS AND DISCUSSION

### Phenotypic measurements

In this study, we investigated muscle fiber diameter and body weight change during the postnatal muscle development stage. As shown in Figs. S1A–S1C, we found that muscle fiber diameter and body weight were significantly increased ($P < 0.001$, one-way analysis of variance) from 0 d to 7 y, which is consistent with results from previous studies in pigs (*Dwyer, Fletcher & Stickland, 1993*; *Fiedler et al., 1997*). Notably, we found that muscle fiber diameter and body weight growth rate significantly slowed with age ($P < 0.05$) (Figs. 1A and 1B). This age-related decrease of muscle fiber growth rate is likely due to decreases in protein synthesis and increases in protein degradation (*Jackman & Kandarian, 2004*). These phenotypic differences indicated possible underlying intrinsic differences in molecular mechanisms.

### Summary of deep-sequencing data

To investigate miRNA expression changes during pig muscle development, nine small RNA libraries were constructed using deep sequencing. In total, 85.84 million raw reads (with redundancy) were obtained. More than 56.67% reads (ranging from 56.67% to 81.25%) in each library passed the Adapter (ADT) dimmer & length, junk, mRNA, RFam, and Repbase filters and were considered high-quality reads (Fig. S2). Of these high-quality reads (Fig. S3), the majority ($90.35 \pm 1.34\%$, $n = 9$) of the small RNAs were approximately 21–24 nt in length. More than half of the reads were 22 nt in length ($57.03 \pm 7.53\%$, $n = 9$), followed by 23 nt ($16.40 \pm 4.34\%$, $n = 9$), 21 nt ($12.37 \pm 5.26\%$, $n = 9$), and 24 nt ($4.55 \pm 1.49\%$, n = 9), which are typical lengths of Dicer-processed products (*Bartel, 2009*; *Berezikov, 2011*). This result indicates that small RNA sequencing is a reliable approach to generate miRNA reads for further analysis.

### miRNA transcriptome profiles during skeletal muscle development

The high-quality reads (sequences) were compared against the pig genome and divided into three groups (Table 1): porcine known miRNAs, porcine conserved miRNAs, and
**Table 1  Porcine and conserved miRNAs detected in nine sRNA libraries.**

| Group (number of pre-miRNA/miRNA) | E90 | 0 d | 30 d | 180 d-1 | 180 d-2 | 180 d-3 | 7 y-1 | 7 y-2 | 7 y-3 |
|---|---|---|---|---|---|---|---|---|---|
| Porcine known miRNAs | 229/379 | 221/372 | 213/340 | 212/321 | 205/312 | 205/322 | 186/260 | 175/233 | 186/272 |
| Porcine conserved miRNAs | 60/83 | 55/78 | 47/67 | 47/62 | 41/55 | 37/49 | 25/28 | 24/28 | 31/37 |
| Porcine putative new miRNAs | 84/94 | 121/140 | 91/106 | 121/145 | 112/135 | 107/122 | 52/60 | 33/40 | 79/94 |

porcine putative new miRNAs. Of the porcine known miRNAs, 404 corresponded to 233 porcine known pre-miRNAs in miRBase 19.0 (http://www.mirbase.org/) and could be mapped to the pig genome. Specifically, 286 miRNAs were known, but 118 were not previously identified porcine miRNAs (Table S2). Of the porcine conserved miRNAs, 101 corresponded to 71 other known mammalian species except for porcine pre-miRNAs in miRBase 19.0, which were further mapped to the pig genome (Table S3). Of the porcine putative new miRNAs, 431 corresponded to 325 candidate pre-miRNAs that were not mapped to mammalian pre-miRNAs in miRBase 19.0, but predicted RNA hairpins derived from the pig genome, and the extended genome sequences form hairpins, which were labeled PN (Table S4). Therefore, 629 pre-miRNAs that encode 936 mature miRNAs were identified. Because there are distinct pre-miRNAs and genomic loci that code identical mature sequences, 903 unique miRNAs were finally identified (Table S5).

To uncover the possible roles of these abundant miRNAs in each stage, we ranked the miRNAs by expression level. The profile of miRNA expression abundance was different across different development stages. As shown in Fig. S4, the most abundant miRNAs (top 10 unique miRNAs) account for more than 50.58% counts of all 903 unique miRNAs. Particularly, the counts of top 10 miRNAs occupied about 90% of the total counts in the 180-d and 7-y libraries, which is significantly higher than that of the E90, 0-d, and 30-d libraries. Furthermore, the set of the top 10 unique miRNAs over the five muscle development stages corresponded to 16 kinds of unique miRNAs (Fig. 2). Of these miRNAs, six (miR-133a-1/-2-3p, let-7a-1/-2-5p, miR-27b-3p, miR-26a-5p, miR-1-3p, and let-7f-1/-2-5p) were shared by all five stages and were closely related to myogenesis, cell growth, myocyte proliferation, and cell apoptosis. For example, miR-133a is a muscle-enriched miRNA that inhibits proliferation of progenitor cells and promotes myogenesis by targeting SRF (*McCarthy & Esser, 2007*). miR-26a promotes myogenesis in C2C12 cells via post-transcriptionally repressing Ezh2, which is a known suppressor of skeletal muscle cell differentiation (*Wong & Tellam, 2008*). These two miRNAs are most abundant during the fast-growing stage of pig muscle. Moreover, miR-27a/b, a potential regulator of myogenesis, could induce skeletal muscle hypertrophy by down-regulating myostatin, an inhibitor of myogenesis (*Huang et al., 2012*; *Sharma et al., 2014*) and miR-27b inhibition leads to more proliferation and delays the onset of differentiation (*Crist et al., 2009*). miR-1, a muscle-specific microRNA, promotes cell apoptosis by targeting Bcl-2 (*Tang et al., 2009*), and could target heat shock protein 70 (HSP70) in the development of muscle atrophy (*Kukreti et al., 2013*). Therefore, miR-27b promotes myogenesis and proliferation, whereas miR-1 inhibits these processes and

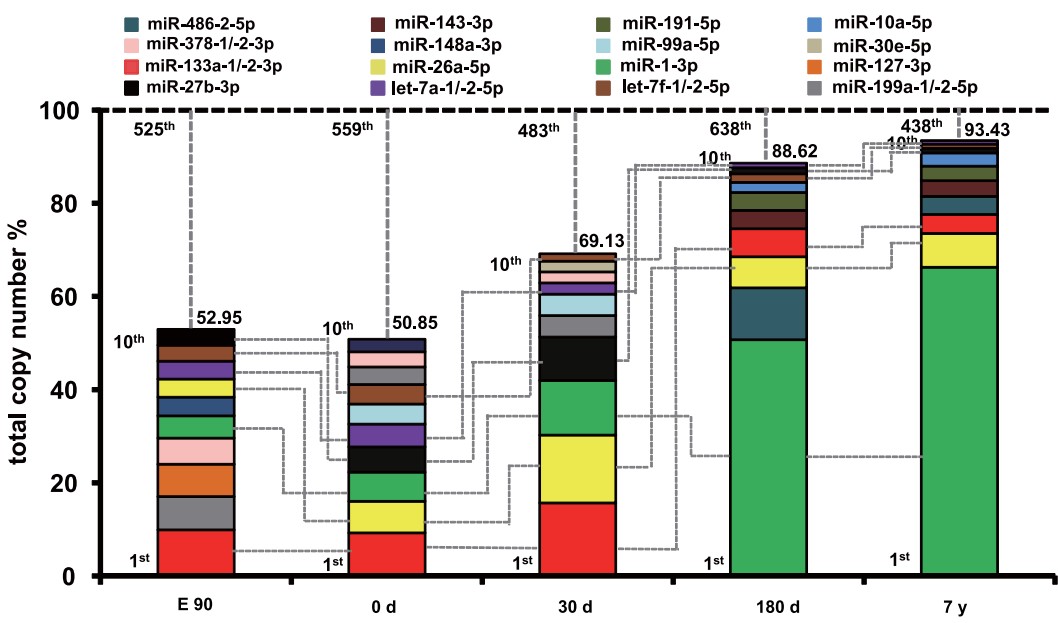

**Figure 2  Top 10 unique miRNAs with the highest expression levels during the five muscle development stages.** Plot of the unique miRNAs versus their total copy number % of all unique miRNAs for each library. The dashed vertical lines represent the cumulative % of the top 10 unique miRNAs in total counts of all unique miRNAs. The 6 miRNAs that are present in the top 10 miRNAs in all libraries are connected by lines.

induces apoptosis. miR-27b showed increasing expression levels during development, and miR-1 showed the opposite trend. These results may indicate fluctuation of miRNA regulation during muscle development.

In addition, 10 other miRNAs (miR-378-1/-2-3p, miR-127-3p, miR-191-5p, miR-486-2-5p, miR-143-3p, miR-10a-5p, miR-148a-3p, miR-99a-5p, miR-30e-5p, and miR-199a-1/-2-5p) (Fig. 2) in the set of the top 10 most highly expressed unique miRNAs over the five muscle development stages are related to various cell proliferation, myogenesis, and apoptosis responses. For example, miR-378 promotes myogenesis in pigs through regulation of bone morphogenetic protein 2 (BMP2) and mitogen-activated protein kinase 1 (MAPK1), especially in fiber formation in both the fetal and newborn periods (*Hou et al., 2012*). In our study, miR-378 showed the highest expression level in the E90 stage. Though Hou et al. found that the expression level of miR-378 increased at 65 and 90 dpc and peaked at postnatal day 0, these results both suggested that miRNA-378 was a new candidate miRNA for myogenesis in pigs (*Hou et al., 2012*; *Qin et al., 2013*). These results supports the idea that it has an important role in fetal muscle development. Another miRNA, miR-148a, belong to the top ten expressed miRNAs only in E90 period, which was in line with previous finding that the average abundance of this miRNA before birth was eight times higher than that in postnatal (*Qin et al., 2013*). miR-486 is directly controlled by SRF, myocardin-related transcription factor-A (MRTF-A), and MyoD, and regulates muscle growth and homeostasis (*Small et al., 2010*). Moreover, miR-191 could promote cell proliferation and suppress apoptosis in MGC803 cells (*Shi et al., 2011*).
These miRNAs were within the top 10 expressed miRNAs in 180-d and 7-y pig muscle, but not in relatively early stages, which indicates that they have regulatory roles in muscle function maintenance after sexual maturity. In contrast, miR-199a is a potential regulator of myogenesis through suppression of WNT-signaling factors and regulation of muscle growth and homeostasis (*Alexander et al., 2013*); its expression was higher in the E90, 0-d, and 30-d stages than in the 180-d and 7-y stages, which indicates that miR-199a is an important myogenic regulatory factor in early fetal and newborn muscle development.

Previous study performed by Qin et al. indicated that most of the highly expressed miRNAs in porcine skeletal muscle such as miR-1 and miR-133 will be more functional. Besides the miR-1 and miR133a that we found highly expressed in all the stages (*Mc-Carthy & Esser, 2007*), miR-26a also showed abundant expression (*Huang, Sherman & Lempicki, 2008*; *Huang et al., 2008*). Taken together, these results indicate that highly abundant miRNAs across various development stages might play important roles in maintenance of vital muscular physiological functions and regulation of skeletal muscle development. Nonetheless, these highly expressed miRNAs also showed dynamic performance across different stages, which indicates typical spatial and temporal-specific expression of miRNAs during muscle development. Undoubtedly, dysregulation of such miRNAs would perturb muscular homeostasis and may even initiate pathogenesis (*Eisenberg et al., 2007*; *McCarthy & Esser, 2007*).

## Differentially expressed (DE) miRNA during muscle development stages

To determine if miRNA expression significantly differed between libraries, IDEG6 (*Romualdi et al., 2003*) was employed to normalize calculations between the high-quality sequences in the nine libraries. A unique miRNA was considered differentially expressed when a general chi-squared test produced *P*-values <0.001 (*Li et al., 2011*). By applying this criterion, we identified 505/903 miRNAs (55.92%) that were differentially expressed during the muscle development stages (Table S6). A previous study demonstrated that miRNAs function in a dose-dependent manner (*Carlsbecker et al., 2010*); thus, only the relatively more abundant miRNAs (>1,000 read counts) were included in the subsequent analysis. Therefore, 167 of 505 (33.07%) differentially expressed (DE) miRNAs were retained (>1,000 read counts).

To further characterize variability in miRNA expression profiling, we performed a hierarchical clustering analysis (HCL) based on the DE miRNAs. As shown in Fig. 3, the miRNA expression profiles showed two major clusters: one that included E90, 0-d, and 30-d stages, and one that included 180-d and 7-y stages. This clustering pattern may correspond to the different growth rate of the muscle fibers in different stages (fast-growth stages: E90, 0-d, and 30-d; slow-growth stages: 180-d and 7-y) (*Dwyer, Fletcher & Stickland, 1993*; *Rehfeldt et al., 2000*). This significant variation in miRNA expression may be responsible, at least in part, for the phenotypic differences (Fig. 1A). Meanwhile, three biological replicates at 180-d and 7-y were highly correlated (180-d: average $r = 0.9532$; 7-y: average

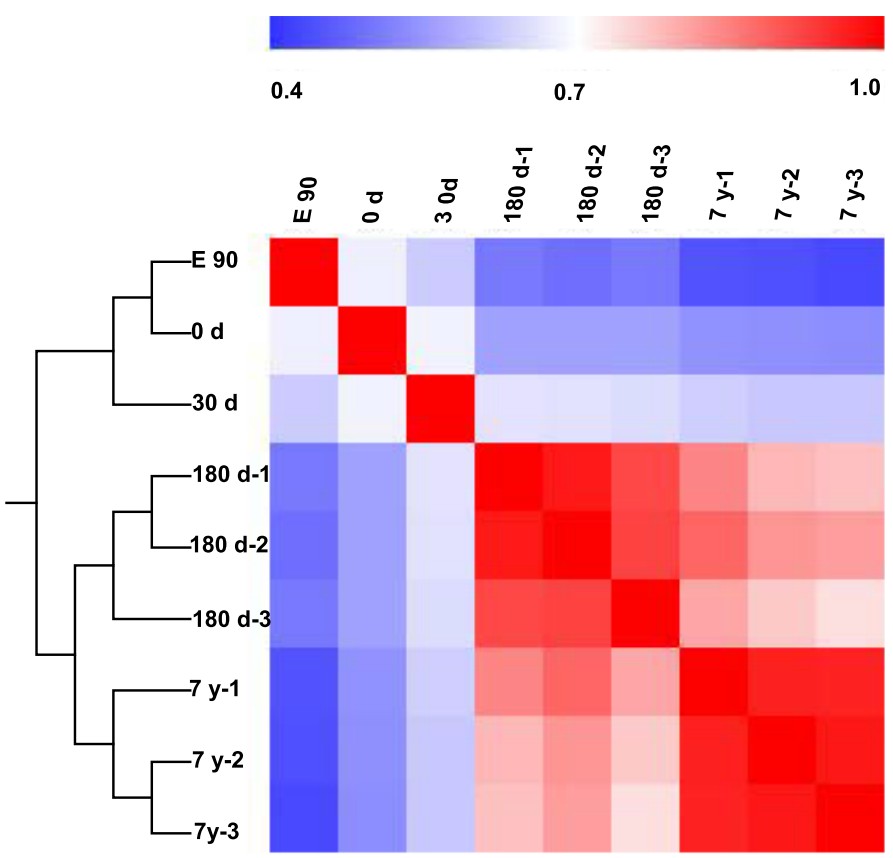

**Figure 3** Hierarchical clustering analysis and heat map matrix of pairwise Spearman correlations of the counts of 167 differentially expressed miRNAs between nine miRNA libraries.

$r = 0.9709$), which indicates high experimental reliability and good reproducibility of replicates (Fig. 3).

Furthermore, nine representative DE miRNAs (miR-133a-5p, miR-181a-1-3p, miR-499-5p, miR-320-3p, miR-24-1-3p, miR-214-3p, let-7g-5p, miR-23a-3p, and miR-10b-3p) were chosen for validation by the stem–loop real time (RT)-PCR-based method using three independent samples. The results were consistent with our sequencing result (Pearson $r = 0.875 \pm 0.119$, $n = 9$; Fig. 4), which indicates that deep sequencing does allow for the successful discovery of miRNAs from pig muscle with high accuracy and efficiency.

### Distinct miRNA expression patterns during muscle development

To visually illustrate the expression pattern of the miRNAs during different development stages, a short time-series expression miner (STEM) analysis was performed for the DE miRNAs. As shown in Fig. 5A, four model profiles had a larger than expected number of genes assigned to the cluster and were significant ($P < 0.05$). For expression pattern 1, expression level decreased across all five stages. For expression pattern 2, expression level increased from E90 to 0 d, then deceased from 0 d to 180 d, and remained stable from 180 d to 7 y. For expression pattern 3, expression level decreased from E90 to 180 d, then deceased from 180 d to 7 y. For expression pattern 4, expression level kept increasing throughout

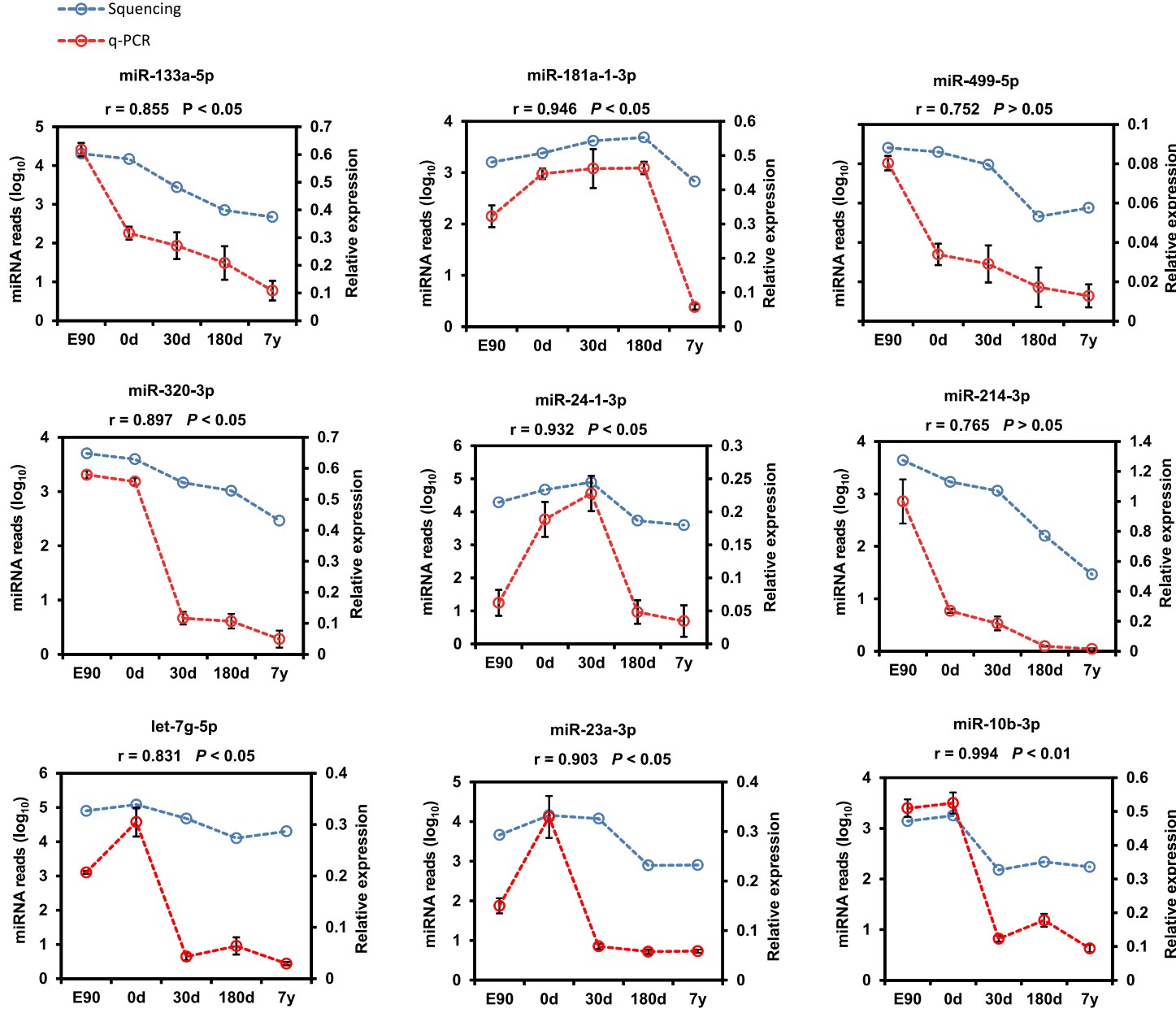

**Figure 4** **Validation of the sequencing data using real-time PCR for nine representative DE miRNAs.** The SPSS software was used to calculate the Pearson correlation coefficient ($r$) and corresponding significance value ($P$).

the five stages. The miRNA list and normalized value of expression by STEM software for each pattern was shown in Table S7. These four profiles may indicate the different roles of corresponding miRNAs; therefore, we performed target prediction and Gene Ontology (GO) analysis. As shown in Fig. 5B, the target genes of highly expressed miRNAs in E90 and 0 d (patterns 1, 2, and 3) were mainly involved in processes related to skeletal muscle development; for example, "cellular component morphogenesis," "embryonic morphogenesis," and "cell morphogenesis involved in differentiation." In mammals,

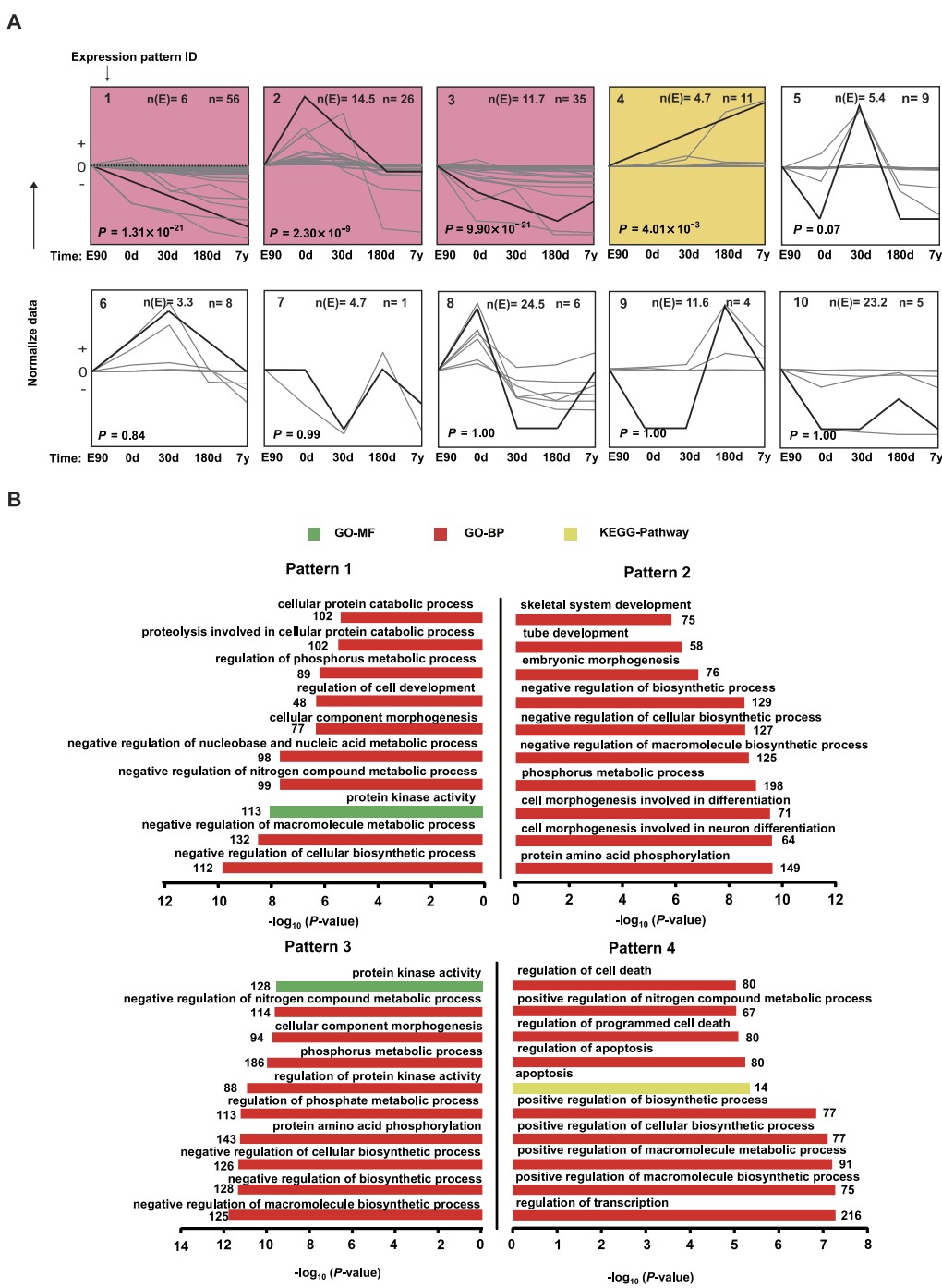

**Figure 5 Expression patterns for differentially expressed miRNAs.** (A) The trends of 167 differentially expressed miRNAs were generally divided into 10 clusters based on their dynamic expression patterns among the five muscle development stages. Numbers in the top left corner indicate the cluster number. Numbers in the top right corner indicate the number of miRNAs in each cluster. The colored clusters indicate clusters that were significantly enriched by miRNAs ($P < 0.05$) (Fisher's exact test). (B) Kyoto Encyclopedia of Genes and Genomes (KEGG) pathways and Gene Ontology-Biological Processes (GO-BP) enriched for target genes of the four significant miRNA clusters. The $P$ values were calculated using Benjamini-corrected modified Fisher's exact test.

muscle mass is mainly determined by the number and size of muscle fibers (*Rehfeldt et al., 2000*). The number of muscle fibers is prenatally determined during primary and secondary muscle fiber formation (about 30–60 d post-gestation) (*Zhao et al., 2003*), whereas postnatal muscle fiber growth is mainly associated with accumulation of muscle-specific proteins (hypertrophy). Therefore, these miRNAs with high expression levels in the embryonic period, which are closely related to cellular morphogenesis and embryonic morphogenesis, may promote differentiation of the muscle fibers in the embryonic period.

In contrast, the target genes of highly expressed miRNAs in the 180-d and 7-y stages (Fig. 5B, pattern 4) were mainly involved in processes such as "apoptosis," "regulation of apoptosis," "regulation of cell death," and "positive regulation of macromolecule biosynthetic process," which are consistent with the results of previous studies (*Dirks & Leeuwenburgh, 2005*). Postnatal growth of skeletal muscle mainly occurs through increases in muscle fiber length and girth and is inhibited in relatively older animals (*Rehfeldt et al., 2000*). Moreover, it has been demonstrated that muscle atrophy induced in middle-aged or older animals is associated with higher amounts of cell apoptosis and death (*Dirks & Leeuwenburgh, 2005*). This may be the primary reason these miRNAs that play vital roles in apoptosis are highly expressed in middle-aged stage. Particularly, miR-1 showed significantly increasing expression level in 180d and 7y stages (Fig. 5A and Table S7). By using both C2C12 myotubes and dex-induced muscular atrophy mouse models, *Kukreti et al. (2013)* indicated that miR-1 is a muscle-specific microRNA and has a role in promoting muscle atrophy. On the other hand, *McCarthy & Esser (2007)* also revealed that miR-1 decreased during mouse skeletal muscle hypertrophy. Our result was consistent with the findings in mouse. Hence, miR-1 could be a potential modulator in regulating porcine postnatal skeletal muscle development.

These results together indicate that there are obviously disparate roles of miRNAs in different development stages of skeletal muscle, such as enhanced levels of miRNAs related to differentiation and morphogenesis in the embryonic stage. Conversely, there were increasing levels of apoptosis-related miRNAs relatively later in muscle development, especially in the middle-aged stage.

## CONCLUSIONS

In this study, we performed a comprehensive investigation of miRNA expression pattern in skeletal muscle across various developmental processes, from embryonic to adult stages, in pigs. We identified a number of DE miRNAs that were associated with porcine muscular development. Through function enrichment analysis of DE miRNAs, we found that the highly expressed miRNAs from E90 to 0 d were mainly involved in macromolecule metabolic, cellular, and embryonic morphogenesis processes, whereas highly expressed miRNAs in the middle-aged stage were mainly involved in apoptosis. Our results indicate that miRNAs act as key regulators of prenatal and postnatal muscle development, and their functions were specific to different stages of skeletal muscle development. In particular, enhanced levels of apoptosis-related miRNAs in middle-aged pigs revealed possible elevated muscle atrophy during this period. However, pigs with more consecutive ages are needed

in further study to uncover the underlying mechanism of refined postnatal pig skeletal muscle development. Our findings will promote further development of pigs as a model organism for human age-related muscle disease research.

### Funding

The authors were supported by grants from the National Special Foundation for Transgenic Species of China (2014ZX0800950B and 2014ZX08006-003), the National Natural Science Foundation of China (31522055, 31530073, 31401073 and 31472081), the Key Project of Sichuan Education Department (15ZA0008/15ZA0003), and the Program for Innovative Research Team of Sichuan Province (2015TD0012). The funders had no role in study design, data collection and analysis, decision to publish, or preparation of the manuscript.

### Grant Disclosures

The following grant information was disclosed by the authors:
National Special Foundation for Transgenic Species of China: 2014ZX0800950B, 2014ZX08006-003.
National Natural Science Foundation of China: 31522055, 31530073, 31401073, 31472081.
Key Project of Sichuan Education Department: 15ZA0008/15ZA0003.
Program for Innovative Research Team of Sichuan Province: 2015TD0012.

### Competing Interests

The authors declare there are no competing interests.

### Author Contributions

- Miaomiao Mai and Long Jin conceived and designed the experiments, performed the experiments, analyzed the data, contributed reagents/materials/analysis tools, wrote the paper, prepared figures and/or tables.
- Shilin Tian, Qianzi Tang, An'an Jiang and Dawei Wang analyzed the data.
- Rui Liu and Wenyao Huang performed the experiments, prepared figures and/or tables.
- Jideng Ma performed the experiments.
- Xun Wang contributed reagents/materials/analysis tools.
- Yaodong Hu contributed reagents/materials/analysis tools, reviewed drafts of the paper.
- Zhi Jiang analyzed the data, reviewed drafts of the paper.
- Mingzhou Li and Xuewei Li conceived and designed the experiments, reviewed drafts of the paper.
- Chaowei Zhou conceived and designed the experiments, wrote the paper, reviewed drafts of the paper.

### Animal Ethics

The following information was supplied relating to ethical approvals (i.e., approving body and any reference numbers):

All animals used in this study were farmed according to the Regulations for the Administration of Affairs Concerning Experimental Animals (Ministry of Science and

Technology, China, revised in June 2004) and approved by the Institutional Animal Care and Use Committee in the College of Animal Science and Technology, Sichuan Agricultural University, Sichuan, China under permit No. DKY-B20121406.

## Data Availability

GEO database accession number GSE50716.

## Supplemental Information

Supplemental information for this article can be found online at http://dx.doi.org/10.7717/peerj.1504#supplemental-information.

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
