# Peer review of "Deciphering the microRNA transcriptome of skeletal muscle during porcine development"

_PeerJ, doi:10.7717/peerj.1504_

## Round 0.1 · original submission · Minor Revisions

· Academic Editor

Minor Revisions

I will be glad to reconsider your paper if you will answer the questions raised by the reviewers.

Reviewer 1 ·

Basic reporting

The manuscript describes a comprehensive investigation of miRNA transcriptomes by RNA sequencing across five porcine muscle development stages including one prenatal stage and four postnatal stages which represented prenatal, natal, weaning, young adult, and middle-aged stages, respectively. The authors identified 404 known porcine miRNAs, 118 novel miRNAs, and 101 miRNAs that are conserved in other mammals and a set of universally abundant miRNAs common to all the development stages. They suggest that this miRNAs may play housekeeping roles in myogenesis. In addition, they describe enhanced expression levels of miRNAs related to differentiation and morphogenesis in the embryonic stage and miRNAs related to apoptosis later in muscle development. Some differentially expressed miRNA were validated by an independent method (qPCR). The manuscript is written in good English, the backgroud and results/discussion section well balanced and the figures sufficiently described.

Experimental design

The experimental design is explained clearly, data well descibed and rigorously conducted and methods sufficiently detailed for reproducibility.

Validity of the findings

The findings appear robust and statistically sound. The data described highlight the importance of miRNA during pig muscle development and represent a source of data for future research.
Some minor points should be addressed:
1) Legend to figures 1 and 4 should give more information on the statistical analysis (test utilized, number of experiments performed, etc.)
2) The sentence in introduction (line 93): “our study will contribute to genetic improvement of meat quality" is not justified by the data. The authors should either explain it or remove it.
3) The authors did not quote the paper from Qin et al, 2013 (Integrative Analysis of Porcine microRNAome during Skeletal Muscle Development by Qin L, Chen Y, Liu X, Ye S, Yu K, et al. (2013) Integrative Analysis of Porcine microRNAome during Skeletal Muscle Development. PLoS ONE 8(9): e72418. doi: 10.1371/journal.pone.0072418), describing a very similar experimental approach. Similarities and differences shold be described and commented.

Comments for the author

No comments

Reviewer 2 ·

Basic reporting

The work of Mai et al. aimed to obtain the microRNA profile of skeletal muscle during porcine development, extending the results of previous analysis, performed mostly during prenatal development, to middle-aged adult muscle when muscle growth is terminated. By including samples from middle aged pigs, authors intended also to identify miRNAs associated with muscle loss. The results of the work might advance the knowledge about miRNAs involved in the regulation of adult myogenesis in pigs, an issue that is still unexplored, therefore they meet the criteria to be published, upon some minor revisions. The purposes of the authors are clear in the introduction, I noticed that the sentence included in lines 74-77 is missing one verb (probably "increase"). In the diagrams of Fig. S1C and D, the p values are not indicated while described in the text: "we found that muscle fiber diameter and body weight were significantly increased (P < 0.001, one-way analysis of variance) from 0 d to 7 180 y". Given that one of the purposes of the authors is to identify miRNAs that are involved in the control of muscle homeostasis, in the "results section " describing the miRNA transcriptome profile, I would suggest to include references that describe the involvement of identified miRNAs in pathways that control hypertrophy and atrophy. For example for miR-27a: Huang et al. 2012, McFarlane et al. 2014. For miR-1: Kukreti et al. 2013, McCarthy & Esser 2007, Elia et al. 2009 etc. In the diagrams of figure 2 the color code is not always useful to identify the associated miRNAs (for example I had difficulties to identify miR-378): I would suggest to enlarge the squares that describe the color code. Authors should compare their results to those obtained in mice (Hamrick et al. 2010) and pigs (Hou et al. 2012, Huang et al 2008) and discuss overlaps and potential contradictions. Given the importance of the middle-aged muscle for the purposes of the authors, I would include the validation of some Differentially Expressed miRNAs that are up-regulated in the muscle obtained from 7 years old pigs, indeed all the 9 miRNAs analyzed in figure 4 are down-regulated in this stage. In figure 5 are reported the expression patterns of differentially regulated miRNAs and I think that, at least for pattern 4, the specific miRNAs should be disclosed to the readers, given that they include miRNAs upregulated in 7 years-old muscles and are involved in the regulation of pathways that control muscle homeostasis, an issue of particular interest in this work.

Experimental design

No comments.

Validity of the findings

Data are robust and findings are interesting, nevertheless I think that it will be important to indicate the miRNAs that are upregulated in middle-aged muscles (pattern 4 of Figure 5). In the conclusion section, authors state that "enhanced levels of apoptosis-related miRNAs in middle-aged pigs revealed possible elevated muscle atrophy during this period" (lanes 328-329), but this statement is not supported by the findings, indeed the average cross-sectional area of the fibers (Fig. S1) do not indicate atrophy.

---

## Round 0.2 · accepted · Accept

· Academic Editor

Accept

The manuscript is now ready for publication on PeerJ.